# AN EMPIRICAL STUDY OF MULTIPLE MASKING IN MASKED AUTOENCODER

## ABSTRACT

The performance of masked autoencoders hinges significantly on masking, prompting considerable efforts towards devising superior masking strategies. However, these strategies mask only once and employ masking directly on the input image. Afterward, inspired by the flexibility of masking, subsequent works introduce two rounds of masking. Unfortunately, all initiatives primarily focus on enhancing model performance, lacking an in-depth and systematical understanding of multiple masking for masked autoencoder. To bridge this gap, this work introduce a masked framework with multiple masking stages, termed Conditional MAE, where subsequent maskings are conditioned on previous unmasked representations, enabling a more flexible masking process in masked image modeling. By doing so, our study sheds light on how multiple masking affects the optimization in training and performance of pretrained models, *e.g.*, introducing more locality to models, and summarizes several takeaways from our findings. Finally, we empirically evaluate the performance of our best-performing model (Conditional-MAE) with that of MAE in three folds including transfer learning, robustness, and scalability, demonstrating the effectiveness of our multiple masking strategy. We also follow our takeaways and show the generalizability to other heterogeneous networks including SimMIM and ConvNeXt V2. We hope our findings will inspire further work in the field and release the code at `https://anonymous.4open.science/r/conditional-mae-512C`.

## 1 INTRODUCTION

Masked autoencoder (MAE) (He et al., 2021) has recently emerged prominently in the field of self-supervised learning (Bao et al., 2021; He et al., 2021; Chen et al., 2021). One of the most representative work, MAE, which partitions an image into visible patches and masked ones, and predict the masked patches from visible ones in RGB space, has gained vast attention from community.

A crucial element of the masked autoencoder is masking, *e.g.*, how (where) to mask and how much to mask, which directly impacts model's performance. Thus, follow-up work develops various masking strategies categorized into four different types by a recent survey (Li et al., 2023b) including Hard Sampling (Kakogeorgiou et al., 2022; Li et al., 2022a; Wang et al., 2023a; Hou et al., 2022; Wu & Mo, 2022), *e.g.*, guided by attention (Kakogeorgiou et al., 2022), Mixture (Chen et al., 2023b; Liu et al., 2022a; Zhang & Shen, 2022), *e.g.*, by mixing different images (Chen et al., 2023b; Liu et al., 2022a), Adversarial (Shi et al., 2022; Chen et al., 2023a), *e.g.*, by introducing adversarial learning (Shi et al., 2022), and Contextual Masking (Li et al., 2022b; Chen et al., 2022a) *e.g.*, using local window (Chen et al., 2022a). Basically, these works mostly mask once and mask only on the input image and focus on how to further improve the performance.

Intuitively, masking is a flexible operation that can be performed at different stages (*e.g.*, the input image and different levels of representations) and with different ratios. Following this line, UnMAE (Li et al., 2022b) introduces two rounds of masking but still perform on the input image. VideoMAE v2 (Wang et al., 2023b) introduces dual masking but primarily focuses on reducing computational costs. $A^2$MIM (Li et al., 2023a) proposes to mask intermediate features from PatchEmbed layer following MAE (He et al., 2021). Though these initiatives have effectively reduced computational costs or enhanced model performance, these efforts have gone only so far, lacking an in-depth and systematical analysis of multiple rounds of masking for masked autoencoder. Hence, a question

naturally arises: *How does multiple masking impact the optimization of the masked autoencoder in both training and performance?*

To answer this question, this work presents a framework called Conditional MAE, which aims to explore the impact of multiple rounds of masking in the training process and performance. In Conditional MAE, subsequent maskings are conditioned on previous unmasked representations, enabling more flexible masking on different granularities of inputs. Based on it, we progressively conduct a thorough empirical study about multiple masking to address three critical questions: 1) *where to mask*, 2) *how much to mask*, and 3) *what's the impact?* In our experiments, we investigate one, two, and three-shot masking [1], where each round of masking is considered a shot. Our results highlight several key takeaways from each shot:

• In the one-shot case, we find that masking at the beginning is always beneficial for task performance. Moreover, it is critical to find a suitable mask ratio. Generally, though the model size is different, *e.g.*, ViT-S and ViT-B, 75% mask ratio is firstly recommended.

• In the two-shot case, building on the best one-shot setting, increasing the interval of two-shot masking with a large ratio followed by a small ratio is helpful for fine-tuning. Additionally, our experiments strongly suggest that there may not exist a positive relationship between linear probing and fine-tuning. Finally, the second masking brings locality bias into the model and helps capture low-level features, especially for finer-grained classification

• In the three-shot case, we find that using a greedy-like masking selection strategy, which uses the best two-shot setting as a starting point, is superior to other three-shot strategies. Simultaneously, the third masking brings more locality into models than two-shot case.

Based on the above results of our empirical experiments, we select the best-performing model (Conditional-MAE) and evaluate its transferability to downstream tasks, including image classification, object detection, and semantic segmentation. We also verify its robustness to noisy inputs, *e.g.*, random occlusion and shuffling, and empirically demonstrate its scalability. Besides, we follow our takeaways and evaluate the generalizability to other network architectures including SimMIM (Xie et al., 2022) and ConvNeXt V2 (Woo et al., 2023).

Note that in this work, we are not to propose a state-of-the-art method, but to enhance both the understanding and performance of MAE by exploring the potential of masking and to inspire future work. Our contributions are three-fold:

• Building on our proposed flexible framework, *i.e.*, Conditional MAE, our primary contribution lies in the first in-depth and comprehensive analysis of how multiple masking influences model optimization from the various aspects including performance comparison, loss, representation, attention map, *etc*.

• Through extensive empirical experiments on multiple masking, we provide several key takeaways from each shot as shown above. More importantly, we observe a key phenomenon that multiple masking is capable of introducing locality bias to models.

• We demonstrate the superiority of our Conditional-MAE over MAE in downstream transfer, robustness against occlusion and shuffling, and scalability. We also show the generalizability to other network architectures.

## 2 CONDITIONAL MAE

### 2.1 PRELIMINARIES

Given an image, MAE first partitions it into $N$ patches $P = \{P^1, P^2, \ldots, P^N\}$ that are randomly categorized into two parts, *i.e.*, visible patches $P_v = \{P_v^1, P_v^2, \ldots, P_v^{N_1}\}$ and masked patches $P_m = \{P_m^1, P_m^2, \ldots, P_m^{N_2}\}$, with a pre-define ratio $\eta_1$ ($N_2 = \eta_1 * N$ and $N_1 + N_2 = N$). Then, $P_v$ are feed into *Encoder* that outputs corresponding patch representations $Z_v = \{z_v^1, z_v^2, \ldots, z_v^{N_1}\}$. Finally, $Z_v$ along with learnable mask token [MASK] [2] are sent into *Decoder* to predict masked

---

[1]Note that we do not study more shots as it is inferior to three-shot masking in our preliminary experiments.

[2]We omit the operation of adding position embedding for description convenience.

Figure 1: An overview of our Conditional MAE. For convenience, we follow MAE (He et al., 2021) and use random masking for each shot. $N_1$, $N_3$, and $N_5$ indicate the number of unmasked patches or representations. In Sec 3.3, we also transfer our conditional framework to other model structures. It is worth mentioning that we do not alter model structures.

patches in RGB space. $P_m$ is served as the supervision signal. The whole process is formulated as:

$$Z_v = Encoder(P_v)\,, \tag{1}$$

$$\widehat{P}_m = Decoder(Z_v,\ [MASK])\,, \tag{2}$$

$$\mathcal{L} = MSE(\widehat{P}_m,\ P_m)\,, \tag{3}$$

where $MSE$ is the mean square error loss function.

## 2.2 CONDITIONAL MAE

Our Conditional MAE is derived from MAE and able to perform multiple shots masking on MAE as shown in Fig 1. We take *two-shot masking* for example to elaborate why we call it Conditional MAE. The first masking is implemented on RGB space with a pre-defined mask ratio $\eta_1$ on image patches, which is what MAE does. Afterward, the second masking is *conditioned* on previous unmasked representations on a given layer of the encoder, *e.g.*, $j$. Thus, for visible patch representations $Z_v^{j^*}$ (output from the $j^*$-th layer of the encoder, $j^* = j - 1$), Conditional MAE mask part of them with another pre-defined masking ratio $\eta_2$. We denote the left visible patch representations as $Y_v^{j^*} = \{y_v^1, y_v^2, \ldots, y_v^{N_3}\}$ and the masked patch representations as $Y_m^{j^*} = \{y_m^1, y_m^2, \ldots, y_m^{N_4}\}$ ($N_3 + N_4 = N_1$ and $N_4 = \eta_2 * N_1$). Additionally, we collect the visible patches corresponding to $Y_m^{j^*}$ from $P_v$, denote them as $P_m^{j^*} = \{P_m^1, P_m^2, \ldots, P_m^{N_4}\}$, and merge them with $P_m$ as $\{P_m,\ P_m^{j^*}\}$ ($\|\{P_m,\ P_m^{j^*}\}\| = N_2 + N_4$) as our new reconstruction target. Therefore, for two-shot masking, the whole process can be formulated as:

$$Z_v^{j^*} = Encoder_{0 \to j*}(P_v)\,, \tag{4}$$

$$Y_v^{j^*},\ Y_m^{j^*} = Mask(Z_v^{j^*},\ \eta_2)\,, \tag{5}$$

$$Z_v = Encoder_{j \to 11}(Y_v^{j^*})\,, \tag{6}$$

$$\widehat{P}_m = Decoder(Z_v,\ [MASK])\,, \tag{7}$$

$$\mathcal{L} = MSE(\widehat{P}_m,\ \{P_m,\ P_m^{j^*}\})\,, \tag{8}$$

where $Encoder_{0 \to j^*}$ means that the input passes through 0-th layer of the encoder and is outputted from $j^*$-th layer.

Compared with MAE, due to the $Mask$ function, the main discrepancies lie in Eq (6) and Eq (8). We need to reconstruct two targets, *i.e.*, $P_m$ and $P_m^{j^*}$, with *less* visible patch representations. Note that this process cannot be bridged by increasing mask ratio $\eta_1$ of MAE to remove more visible patches. We explain it below. For $P_m$, similar to MAE, it has never been seen by the encoder and thereby we need *infer* it via visible patch representations $Y_v^{j^*}$. For $P_m^{j^*}$, it has been seen by partial encoder (*i.e.*, layers before $j$), resulting in its information involved in $Y_v^{j^*}$ via attention-manner interaction between $Y_v^{j^*}$ and $Y_m^{j^*}$ before $j$-th layer. We reconstruct the patches $P_m^{j^*}$ primarily *conditioned* on the "borrowed" information involved in $Y_v^{j^*}$ via the interaction above. This is easily generalized to multiple shots. Particularly, in the two-shot showcase, if $j$ is set to 0 or $\eta_2$ is 0, Conditional MAE is

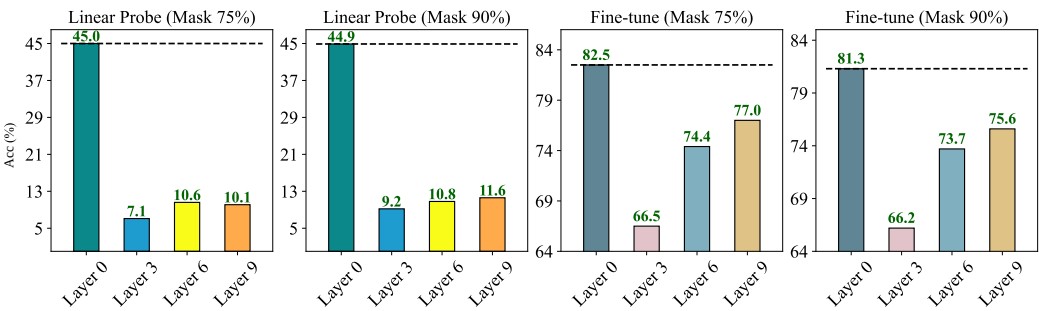

Figure 2: Results of one-shot masking on ViT-S/16.

reduced to MAE. if $\eta_1$ (the first mask ratio) is 0, our Conditional MAE is still established with only reconstruction of $P_m$ removed.

## 3 EXPERIMENT

### 3.1 MULTIPLE SHOTS MASKING

In our study, we investigate the Conditional MAE in three different settings by pretraining on ImageNet100: one-shot masking, two-shot masking, and three-shot masking. We do not explore settings with more shots, as preliminary experiments have shown them to be inferior to three-shot. For ease of description, we denote the three mask ratios as $\eta_1, \eta_2, \eta_3$, and the corresponding layer indexes as $i, j, k$, respectively, where masking is applied before inputting. Considering our Conditional MAE is derived from MAE, we fix $i = 0$ to match with MAE. Through exhaustive experiments conducted below, we aim to address three key questions: *where to mask*, *how much to mask*, and *what is the impact?* For training details, please refer to the Appendix A.1.

#### 3.1.1 ONE-SHOT MASKING

In the one-shot setting, we only mask patch tokens in the encoder once, allowing us to examine the impact of different mask positions and mask ratios on encoder performance. Specifically, for mask positions, we consider four positions at equal intervals: the 0-th, 3-th, 6-th, and 9-th layer of encoder blocks, denoted as $(i, j, k) = (0, 0/3/6/9, 0)$. We exclude the 12-th layer as it cause a denoise autoencoder to degenerate into a vanilla one. Regarding mask ratios, we carefully select two representative ratios used in MAE (He et al., 2021), namely 0.75 and 0.9, denoted as $(\eta_1, \eta_2, \eta_3) = (0, 0.75/0.9, 0)$ [3]. The reasons are two-fold: 0.75 is widely used in MAE; For 0.9, previous work (Riquelme et al., 2021) has shown that even using 10% patch features can still yield competitive performance in visual recognition. The results on ViT-S/16 are illustrated in Fig 2.

*It has been observed that masking at the beginning position ($j = 0$) is beneficial for both linear probing and fine-tuning.* Conversely, we also notice a significant drop in performance for linear probing when masking is applied at the other positions. This implies that the representations encoded by the fixed encoder at $j = 0$ are relatively more distinguishable while other encoders learn comparatively less knowledge compared to the encoder at $j = 0$, which could be attributed to the information leakage from attention interaction. To support this speculation, we visualize the training loss curves of pretraining and linear probing and t-SNE of output representation in Appendix A.2.1.

Finally, to investigate the impact of mask ratio on models of different sizes, we also conduct experiments on ViT-B/16 and present the results in Tab 1. Interestingly, we observe that a mask ratio of 0.75 enhances the performance of ViT-B/16 compared to a mask ratio of 0.9, which is similar to ViT-S/16. Moreover, our results are consistent with MAE (He et al., 2021) trained on ImageNet1k (Russakovsky et al., 2015) with best mask ratio 75%.

Table 1: Comparisons on ViT-S/16 and ViT-B/16 with different mask ratio.

| Model Size | Mask Ratio | Linear Probe | Fine-tune |
|---|---|---|---|
| ViT-S/16 | 0.75 | **45.0** | **82.5** |
| | 0.90 | 44.9 | 81.3 |
| ViT-B/16 | 0.75 | **62.9** | **86.9** |
| | 0.90 | 57.9 | 85.6 |

---

[3] We set $\eta_1$ to 0 as its layer index $i = 0$ is fixed as described at beginning while our mask position should be flexible.

***Takeaways.*** *For one-shot masking, we summarize two useful tips: ① Masking at the beginning is always beneficial for task performance; ② Finding a suitable mask ratio is critical. Generally, though the model size is different,* e.g.*, ViT-S and ViT-B, a 75% mask ratio is firstly recommended.*

***Remark.*** Though mask ratio ablation has been explored in MAE paper, our focus on the one-shot masking aims to delve deeper into the specifics of how different masking positions and ratios collectively influence the model's performance. Moreover, it serves as the basis for the subsequent two-shot masking.

### 3.1.2 TWO-SHOT MASKING

Two-shot masking means we can mask twice in the encoder. For convenience, we use $L(i, j)$ ($k$ is omitted) to indicate that we mask the $i$-th and $j$-th **Layers** ($i = 0$ and $i < j < 12$). We use $(\eta_1, \eta_2)$ ($\eta_3$ is omitted) to denote the mask ratio of two-shot masking. By combining them, $L(i, j; \eta_1, \eta_2)$ means that we mask the $i$-th layer with mask ratio $\eta_1$ and mask the $j$-th layer with mask ratio $\eta_2$. To fully explore model capabilities, we follow the conclusion from one-shot and first mask patch tokens at the beginning also with two representative mask ratios (0.75 and 0.9). *Thus, it is critical to figure out where the second masking should be and how much it should mask.* The experiments on ViT-S/16 are shown in Fig 3. The dashed line denotes the one-shot baseline (*i.e.*, MAE) with masking ratios of 0.75. It is worth mentioning that compared to standard MAE, the adopted two-shot masking is capable of reducing the computational cost as it introduces additional masking to remove more tokens in the forward pass, that is, fewer tokens are involved in the forward calculation [4]. Since it is not our primary focus, we would not highlight the advantage in this aspect.

For $\eta_1 = 0.75$, we ablate five combinations of mask layers for two-shot masking. Three involve an equal interval for the second masking layer indexes following the one-shot masking scheme: $L(0, 3)$, $L(0, 6)$, and $L(0, 9)$; Two are continuous combinations: $L(0, 10)$ and $L(0, 11)$ [5]. We initially set a larger mask ratio of $\eta_2$ (0.5). Considering that the performance is inferior to the baseline in both linear probing and fine-tuning, we replace $\eta_2 = 0.5$ with three relatively smaller ones containing 0.25, 0.15, and 0.1. As shown in Fig 3 (a), the performance of two-shot masking is inferior to the baseline

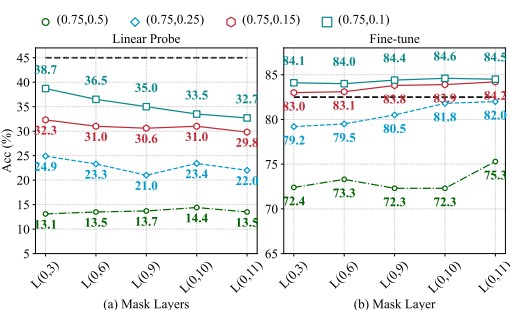

Figure 3: Results of two-shot masking on ViT-S/16. The dashed line is the one-shot baseline (MAE).

for linear probing. However, as opposed to linear probing, one can see in Fig 3 (b) that our two-shot masking shows potential to outperform the baseline in fine-tuning: An apparent trend for fine-tuning is that the second masking performed at the last several layers (*i.e.*, increasing the interval of two-shot masking) with a smaller $\eta_2$ leads to significant improvement compared to baseline, especially at $L(0, 10)$ [6]. *The contradictory experiment results imply that there may not exist a positive correlation between linear probing and fine-tuning.* Hence, following (Woo et al., 2023), we would pay more attention to fine-tuning because of its practical relevance in transfer learning. Two-shot results of $\eta_1 = 0.9$ is in Appendix A.3.1

Given the superior improvement, a question arises: **what two-shot masking brings to the encoder?** We dive deep into two-shot masking and analyze its layer representation and attention map.

---

[4]For example, on 4 A800, our Conditional MAE (ViT-Large with two-shot L(0,10;0.75.0.1)) is around 650ms/iteration while MAE is 670ms/iteration when input size is 224x224 and batch size is 256. Moreover, Conditional MAE saves around 1G GPU memory compared to MAE. If batch size and input size is larger and training time is longer, the saved memory and time will be considerably impressive.

[5]In our preliminary experiments, $L(0, 9)$ performs the best in fine-tuning among these three combinations. To provide a more comprehensive analysis, we include $L(0, 10)$ and $L(0, 11)$. We do not include $L(0, 8)$ as it performs worse than $L(0, 9)$.

[6]We also verify this characteristic in large-scale ImageNet1K and ViT-Large in Sec 3.4. Moreover, we compare the performance of our best two-shot masking $L(0, 10; 0.75, 0.1)$ with one-shot $L(0; 0.775)$ where they retain the same number of patch tokens. The one-shot perfromance (83.2% Acc) is inferior to our two-shot (84.6%). Similarly, we compare two-shot masking $L(0, 10; 0.9, 0.05)$ with one-shot $L(0; 0.905)$. Our two-shot (82.1%) also outperforms one-shot (81.1%) whose performance is even inferior to that of $L(0; 0.9)$ (81.2%).

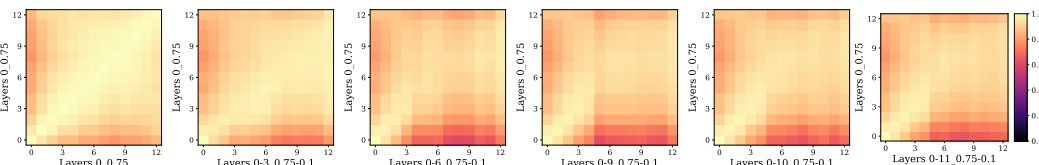

Figure 4: Layer representation similarity between pretrained two-shot masking model and baseline.

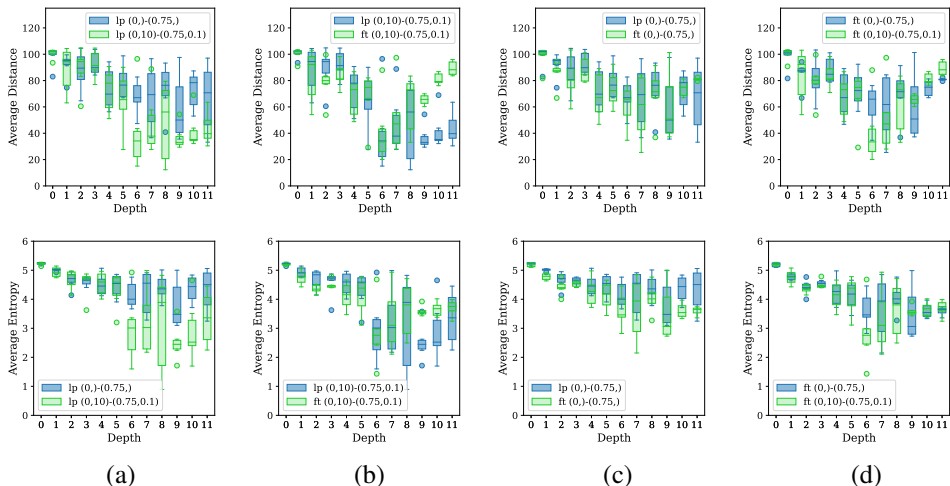

Figure 5: Comparison of two-shot masking $L(0, 10; 0.75, 0.1)$ and baseline model $L(0; 0.75)$ on attention distance and attention entropy before/after fine-tuning. The lp means pretrained model. The ft means fine-tuned models.

***Layer Representation Analyses.*** We first leverage Centered Kernel Alignment (CKA) (Cortes et al., 2012; Nguyen et al., 2020) to analyze the layer representation similarity across pretrained models [7]. As shown in Fig 4, we visualize the layer representation similarity between several two-shot masking pre-trained models and baseline $(0, 0.75)$ as heatmaps. It is seen an increasing discrepancy between the representations of two-shot models and that of baseline, especially between the high layer of two-shot models and shallow layer of baseline. This implies that the second masking introduce certain bias into pretrained models, rendering the representations varying from that of baselines [8].

***Attention Map Analyses.*** We then analyze the attention maps that reveal the behaviors for aggregating information in the attention mechanism of ViTs. Following (Wang et al., 2023c) we use two metrics, *i.e.*, attention distance and attention entropy [9], to analyze two-shot masking and baseline models. We pick $L(0, 10; 0.75, 0.1)$ as it performs best and illustrate its attention distance and entropy variation before/after fine-tuning and compare with that of baseline $L(0; 0.75)$ in Fig 5. We see that the second masking decreases the attention distance and entropy to some extent during pretraining in Fig 5 (a), bringing locality inductive bias into model and thereby rendering the representations varying from that of baselines. From the view of reconstruction, we conjecture such adjustment is because the second masking requires the unmasked patches to recover their parallel neighbor (masked ones) of a forward. In Fig 5 (b) and (c), compared to pretraining, we find that fine-tuning actually demonstrates similar behavior. Specifically, for lower layers, both $L(0, 10; 0.75, 0.1)$ and $L(0; 0.75)$ decrease their lower and upper bounds of attention distance during fine-tuning compared to pretraining. For higher layers, both models increase their lower bound of attention distance. Finally we compare the attention distance and entropy between the two models after fine-tuning in Fig 5 (d) to figure out what makes $L(0, 10; 0.75, 0.1)$ have potential to outperform baseline $L(0; 0.75)$. We see that $L(0, 10; 0.75, 0.1)$

---

[7]CKA computes the normalized similarity in terms of the Hilbert-Schmidt Independence Criterion (HSIC (Song et al., 2012)) between two feature maps or representations.

[8]Note that the disparity in the heatmap does not necessarily imply whether the learned representation is advantageous or detrimental. It reflects how the representation learned by our two-shot masking model varies from that of the baseline.

[9]The attention distance reveals how much local *vs.* global information is aggregated, and a lower distance means each token focuses more on neighbor tokens. The attention entropy reveals the concentration of the attention distribution, and lower entropy means each token attends to fewer tokens. We refer the reader of interest to (Wang et al., 2023c) for detailed formula

Figure 6: Visualization of reversed attentions (showing how much information a second-masked patch sends to others) in layer 9 of models. Top: single masking model $L(0; 0.75)$ (vanilla MAE). Bottom: two-shot masking model $L(0, 10; 0.75, 0.1)$. By comparing every map pair, one can see that these second-masked tokens tend to send and store the information to their neighbors just prior to being masked. Especially in the centered highlighted part, $L(0, 10; 0.75, 0.1)$ tends to be more compact and localized.

has similar attention distance and entropy in high layers while more concentrated and lower attention distance and entropy in low and middle layers. We attribute it to locality inductive bias brought by the second masking that captures better low-level features. Similar observations can be found in other two-shot model variants ($\eta_1 = 0.75$ and $0.9$) which we put in Appendix A.3.2.

***Information Leakage and Locality.*** In the two-shot setting, the second masked patches have been seen by previous layers, potentially resulting in information leakage. However, it's important to note that this leakage does not cause a trivial solution as the presence of $\eta_1$ and its substantial gap in magnitude compared to $\eta_2$ necessitates the model to acquire the ability to infer the masked patch in the first masking. In contrast, the presence of the second masking necessitates that patches that interacted in previous layers must recover their corresponding masked neighbors in the forward pass. As a result, the model needs to dedicate a portion of its capacity to learn how to infer local neighbors. This would introduce a certain degree of locality bias. To illustrate this, we visualize the reversed attention (Ding et al., 2023) of pretrained model $L(0, 10; 0.75, 0.1)$ as shown in Fig 6 (bottom), containing the information flow of second masking, *i.e.*, how much information a second-masked patch sends to other. It clearly demonstrates that the attention head retains object-related local information. In this way, the information leakage is controllable, and information of the second-masked patch flows and is stored in the neighboring patches, to be reconstructed after the second masking. Also, compared with single masking in Fig 6 (top), the locality of the attention head is enhanced, potentially benefiting some downstream tasks that require low-level or local representations (Jiang et al., 2022).

***Potential Application.*** Intuitively, the derived locality of two-shot masking allows models to capture nuanced, locally fine-grained characteristics, thereby discern subtle distinctions between close classes. To prove this, we conduct fine-grained classification on three widely-used fine-grained datasets including Flower102 (Nilsback & Zisserman, 2008), Stanford Dog (Khosla et al., 2011), and CUB-200 (Wah et al., 2011) using ViT-S, and compare the results with that of ImageNet100 (generic classification) in Tab 2. We find $L(0, 10; 0.75, 0.1)$ obtains more enhancement than $L(0; 0.75)$ in fine-grained classification.

Additionally, a subtle and interesting phenomenon is captured during our experiments. We take $L(0, 10; 0.75, 0.15)$ and $L(0, 10; 0.9, 0.1)$ for example and in Fig 7, the second reconstruction loss (orange) of masked patches (2nd shot) unanimously decreases faster than that of the first (blue) (1st shot). This result indicates the second reconstruction task is

Table 2: Comparisons on fine-grained datasets.

| Dataset | $L(0; 0.75)$ | $L(0, 10; 0.75, 0.1)$ |
|---|---|---|
| ImageNet100 | 82.5 | 84.6 (+2.1) |
| Flower102 | 34.7 | 37.3 (+2.6) |
| Standford Dog | 51.6 | 54.3 (+2.7) |
| CUB-200 | 48.2 | 51.1 (+2.9) |

relatively easier to optimize than the first. To some extent, using the same loss weights for them is unreasonable and wastes model's capability. Hence, intuitively, we adopt their mask ratios as their new loss weights during training to force the model to concentrate more on the first reconstruction task. *In Tab 3, we find that this adjustment significantly improves the performance of linear probing but has limited enhancement on fine-tuning.* Since our focus is primarily on the performance of finetuning, we did not adopt this strategy in our experiments and leave it as a potential avenue for future exploration.

Finally, we apply our findings in ViT-S/16 on ViT-B/16, hoping to further improve its performance as well. Since the performance of $\eta_1 = 0.9$ for ViT-B/16 in Tab 1 is inferior to that of $\eta_1 = 0.75$, we focus primarily on $\eta_1 = 0.75$ for ViT-B/16 in the experiment. Specifically, we employ the three best two-shot settings of finetuning performance of ViT-S/16 on ViT-B shown in Tab 4 and compare the results with MAE. Our two-shot masking strategy unanimously outperforms MAE. And among them, $L(0, 10; 0.75, 0.1)$ performs best, which also performs best for ViT-S/16.

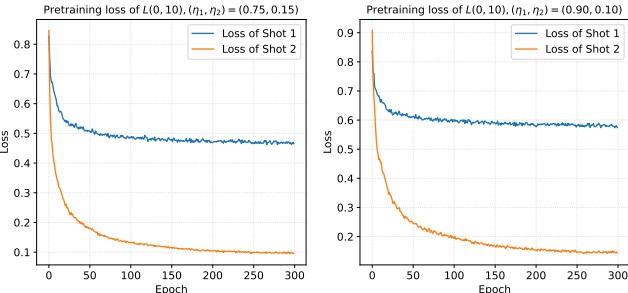

| $\eta_1, \eta_2$ | $w_1, w_2$ | LP | FT |
|---|---|---|---|
| 0.75, 0.15 | 0.5, 0.5 | 31.0 | 83.9 |
| | 0.75, 0.15 | **35.2** | 83.9 |
| 0.9, 0.1 | 0.5, 0.5 | 35.5 | 81.9 |
| | 0.9, 0.1 | **36.3** | **82.0** |

Table 3: Results of different loss weights

| $i, j$ | $\eta_1, \eta_2$ | FT |
|---|---|---|
| L(0) | 0.75 | 86.88 |
| L(0,10) | 0.75, 0.1 | **87.66** |
| L(0,10) | 0.75, 0.15 | 87.46 |
| L(0,11) | 0.75, 0.1 | 87.26 |

Figure 7: Training loss curves of two-shot.      Table 4: Results on ViT-B/16 compared to MAE

***Takeaways.*** *For two-shot masking, we summarize four useful findings: ① building on one-shot, increasing the interval of two-shot masking with a large $\eta_1$ and a small $\eta_2$ is helpful for fine-tuning in both ViT-S/16 and ViT-B/16, e.g., $L(0, 10)$ in our experiments; ② it strongly suggests that there may not exist a positive relationship between linear probing and fine-tuning; ③ the second masking brings locality bias into model and help capture low-level features, especially for finer-grind classification; ④ adopting a weighted reconstruction loss for different shot is helpful for linear probing.*

**Remark.** Our empirical analysis reveals the impact of two-shot masking on representation and attention maps. Moreover, we believe it would be interesting to explore the multi-stage masking process from an information theory perspective, *e.g.*, examining how the information content evolves at each masking stage, which we plan to address in future work.

### 3.1.3 THREE-SHOT MASKING

We further explore the three-shot masking. Specifically, we leverage a greedy-algorithm-like strategy by using the best two-shot setting $L(0, 10; 0.75, 0.1)$ and add the third masking on the last layer of encoder ($k = 11$) with a small masking ratio $\eta_3 = 0.1$. We verify the effectiveness of our three-shot masking by comparing it with various strategies including "Equal interval", "Prefer front layer", and "Unbalanced interval". Moreover, we also find there do not exist a positive relationship between linear probing and fine-tuning in three-shot masking. For example, $L(0, 10, 11; 0.75, 0.1, 0.1)$ achieves 29.6% Acc in linear probing, inferior to $L(0, 1, 10; 0.75, 0.1, 0.1)$ (31.0%). But $L(0, 10, 11; 0.75, 0.1, 0.1)$ achieves 81.9% Acc in fine-tuning, superior to $L(0, 1, 10; 0.75, 0.1, 0.1)$ (81.8%). Besides, by visualizing the attention distance and entropy and comparing with that of two-shot and one-shot masking, we find the third masking introduces a more prominent locality bias as shown in Fig 20. Similarly, we conduct fine-grained classification in Tab 8 and find that though the model outperforms the baseline but the enhancement is inferior to that of two-shot. Intuitively, we speculate that this would be due to the over-locality introduced by the third shot masking. Due to the limited space, we put all the results in Appendix A.4.

***Takeaways.*** *In three-shot masking, we find that a greedy-like masking strategy is superior over a wide range of strategies. And more prominent locality is brought into models.*

Table 5: Downstream performance of Conditional-MAE compared to MAE. DTD (Cimpoi et al., 2014), CF means CIFAR (Krizhevsky et al., 2009). Tiny indicates TinyImageNet (Le & Yang, 2015)

| Model | Classification | | | | Obj Det | | Sem Seg |
|---|---|---|---|---|---|---|---|
| | DTD | CF10 | CF100 | Tiny | $AP^b$ | $AP^m$ | |
| MAE | 57.9 | 84.5 | 62.5 | 63.4 | 38.9 | 35.1 | 38.3 |
| Conditional-MAE | **59.1** | **85.5** | **63.4** | **64.1** | **39.5** | **35.5** | **38.9** |

### 3.2 TRANSFER LEARNING

To conduct transfer learning in downstream tasks, we compare the best results of one-shot (82.5 Acc@1), two-shot (84.6 Acc@1), and three-shot (Acc@1), among which two-shot performs the best. Hence, we pick up the best two-shot masking ImageNet100 pretrained ViT-B/16 model (Conditional-MAE). To verify its effectiveness in transfer learning, We perform classification on four datasets,

| Model | $L(i, j; \eta_1, \eta_2)$ | FT |
|---|---|---|
| SimMIM (Swin-B) | $L(0; 0.6)$ | 83.8 |
| SimMIM (Two-shot Swin-B) | $L(0, 3; 0.6, 0.1)$ | **84.9** |
| ConvNeXt V2-B | $L(0; 0.6)$ | 80.2 |
| ConvNeXt V2-B (Two-shot) | $L(0, 3; 0.6, 0.1)$ | **81.1** |

Table 6: Comparisons on two-shot variants and baseline.

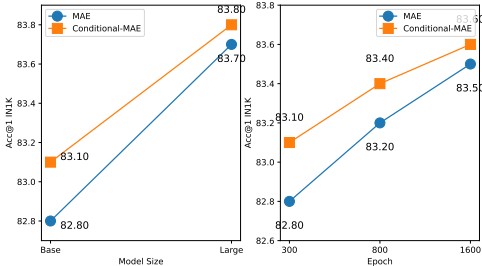

Figure 8: Results of scaling Conditional-MAE on larger model and longer training time.

object detection on COCO (Lin et al., 2014), and semantic segmentation on ADE20K (Zhou et al., 2017; Zhu et al., 2022) following previous works (He et al., 2021; Chen et al., 2022b; Zhou et al., 2021). As shown in Tab 5, Conditional-MAE generally produces better performance than MAE in downstream tasks, showing its great transfer capability. Besides, we consider pretraining Conditional-MAE on larger ImageNet1K with 300 epochs and fine-tune it on downstream task, *e.g.*, semantic segmentation using ADE20K. The Conditional-MAE produces 46.1 for mIoU, superior over MAE (45.8), further verifying the effectiveness of two-shot masking. In addition of transfer learning, we also shows the robustness of Conditional-MAE over MAE against occlusion and shuffling in Appendix A.5.

### 3.3 GENERALIZATION

Besides MAE, we follow our empirical takeaways above and perform two-shot masking on two different structures including SimMIM (Swin Transformer) and ConvNeXt V2 (CNN). In Tab 6, since both SimMIM and ConvNeXt V2 use a hierarchical encoder, here we use $i$ and $j$ to denote different stages. We see that the two-shot variants outperform the baselines, showing great generalization of our two-shot masking to other structures beyond ViT.

### 3.4 SCALABILITY

To verify the scaling capability, we pretrain Conditional-MAE on ImageNet1K (Russakovsky et al., 2015), scaling on large model, *i.e.*, ViT-L, and longer pretraining times, *e.g.*, 1600 epoch. The result is presented in Fig 8 where the left is training with 300 epochs for both models and the right uses ViT-B/16. It is shown that pretraining Conditional-MAE with a longer time and increasing the size of pretrained Conditional-MAE can significantly improve performance, demonstrating promising scaling capability of Conditional-MAE.

## 4 RELATED WORK

**Masked image modeling.** Masked image modeling is the task of predicting the masked part of an image from the visible part. Inspired by masked language modeling in natural language processing, BEiT (Bao et al., 2021) is the first to employ this paradigm in computer vision. PeCo (Dong et al., 2021) further improves the performance of BEiT by involving more semantics in visual tokens. MAE (He et al., 2021) removes the need for a tokenizer (*e.g.*, d-vae (Ramesh et al., 2021) in BEiT) by directly predicting the masked part in RGB space. This greatly simplifies the whole pipeline and improves the model performance simultaneously. CAE (Chen et al., 2022b) adds a regressor between the encoder and decoder to align masked and visible representations in the same representation space. iBOT (Zhou et al., 2021) combines masked image modeling with contrastive learning, showing great potential. (Shi et al., 2022) uses an adversarial objective to consistently improve on state-of-the-art self-supervised learning (SSL) methods. MaskFeat (Wei et al., 2022a) uses Histograms of Oriented Gradients (HOG), a hand-crafted feature descriptor, as reconstruction target. Recently, with more effort devoted to this field, numerous works (Dong et al., 2022a; Gao et al., 2022; Zhang et al., 2022b; Chen et al., 2022c; Kakogeorgiou et al., 2022; Li et al., 2021; El-Nouby et al., 2021; Liu et al., 2022b; Tao et al., 2022; Wei et al., 2022a; Zhang et al., 2022a; Yu et al., 2022; Assran et al., 2022; Fang et al., 2022; Bachmann et al., 2022; Shi et al., 2022; Wei et al., 2022b; Huang et al., 2022a;b; Dong et al.,

2022b) are proposed including BootMAE (Dong et al., 2022a), SdAE (Chen et al., 2022c), MST (Li et al., 2021), SplitMask (El-Nouby et al., 2021), SIM (Tao et al., 2022), *etc*.

**Understanding masked image modeling.** Xie *et al*. shows that masked image modeling brings rich diversity to the self-attention head and pays more attention to locality compared to supervised one (Xie et al., 2021b). Additionally, Xie *et al*. also demonstrates that larger models, more data, and longer training times are beneficial for masked image modeling (Xie et al., 2021a). CAE (Chen et al., 2022b) illustrates its attention map and speculates that masked image modeling cares more about the global including both foreground and background. Kong & Zhang (Kong & Zhang, 2022) point out that masked image modeling brings occlusion invariant to the model representation. Cao *et al*. (Cao et al., 2022) deliver a mathematical understanding of masked image modeling. More recently, Zhu *et al*. (Zhu et al., 2023) speculate that masked image modeling is a part-to-part process: the masked representations are hallucinated from the visible part.

**Masking.** Masking is a key operation in masked image modeling. Trandional masked strategies include random masking used in MAE (He et al., 2021), and block masking used in BEiT (Bao et al., 2021) and CAE (Chen et al., 2022b). Besides, previous works also explore extra masking strategies. MST (Li et al., 2021) masks low-attended patches to enhance the performance without additional cost. AttMask (Kakogeorgiou et al., 2022) further proves the usefulness of masking highly attended portions. AMT (Gui et al., 2022) uses the attention map in the last layer of the vision transformer to guide the masking. SemMAE (Li et al., 2022a) leverages a masking with semantics provided by an additional pretrained model. However, it is worth noticing that almost all of them mask an image just at the beginning and primarily focus on how to further improve the performance. UnMAE (Li et al., 2022b), VideoMAE v2 (Wang et al., 2023b), and A$^2$MIM (Li et al., 2023a) introduces two rounds of masking but these efforts have gone only so far, lacking an in-depth and systematical analysis of multiple rounds of masking for masked autoencoder. In contrast, our work fills this gap and reveals the secret of multiple masking on masked autoencoder's optimization with different masking positions and ratios. We also discuss masking in generation modeling in Appendix A.6.

## 5 CONCLUSION

In this paper, we reveal how multiple masking affects masked autoencoder's optimization in training and performance by using a flexible framework called Conditional MAE. Based on our findings, we summarize several takeaways from each shot and find that multiple masking can bring locality bias to models. We also show the superiority of our best two-shot model Conditional-MAE over MAE in downstream tasks, robustness again occlusion and shuffling, masking generalizability to other heterogeneous architectures, and model scalability, providing sufficient insight for future work.

**Limitation and Broader Impact.** Our study is constrained by limited computational resources. We conducted our experiments using small, base, and large ViT. Therefore, it would be interesting to extend this study to larger models *e.g*., Huge ViT. Our empirical study primarily focuses on the masked autoencoder. There may not exist any negative effects on itself but on how it is used.

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

# A   APPENDIX

## A.1   IMPLEMENTATION DETAILS

**Pretraining.** Similar as MAE, we use the original image without color jittering, gradient clip, or other transformations. For the experiments in multiple shots masking, we conduct pretraining on the ImageNet-100 dataset, which is a subset of ImageNet-1K and contains 135,000 images from 100 random classes with $224 \times 224$ pixels. The batch size is 256, weight decay is 0.05, warmup epochs is 40, and the base learning rate is $1.5e - 4$ following MAE (He et al., 2021). We train each model for 300 epochs equally. For pre-training, we use AdamW as the optimizer.

**Linear Probe.** Following MAE (He et al., 2021), we conduct training of linear probing for 90 epochs with learning rate 0.1 and 1024 batch size. CLS token is used for classification. The LARS optimizer is utilized for linear probing.

**Fine-tuning.** We fine-tune pretrained model for 100 epochs following MAE (He et al., 2021). The weight decay is 0.05 and layer decay is 0.65. We set drop path to 0.1. We search from three base learning rates, $1e - 4$, $5e - 4$, and $1e - 3$. The batch size is 256. We use AdamW as the optimizer with warmup epoch set to 5 and cosine learning rate scheduler. Following MAE (He et al., 2021), we use global pooled representation for classification. In part classification, we set batch size to 16 for all datasets and use two base learning rates $7e - 4$ and $8e - 4$ respectively while maintaining other setting.

**Transfer learning.** In downstream transfer, we use the final pretrained checkpoint to initialize model and then finetune it. For classification, we finetune for 100 epoch using AdamW as optimizer with weight decay 0.05 and layer decay 0.65. Due to the computation limitation, we use $32 \times 32$ for CIFAR100 and CIFAR10 with batch size 512 and set learning rate to $1.5e - 3$. We use $64 \times 64$ for TinyImageNet with batch size 512 and set learning rate to $1.5e - 3$. We use $448 \times 448$ for DTD with batch size 16. We use learning rate $2.5e - 3$. For semantic segmentation, we following CAE (Chen et al., 2022b) The input resolution is $512 \times 512$. The batch size is 16 and the layerwise decay rate is 0.65 and the drop path rate is 0.1. We search from three learning rates, $3e - 4$, $4e - 4$, and $5e - 4$. We conduct fine-tuning for 160K steps. We do not use multi-scale testing. For object detection, we utilize multi-scale training and resize the image with the size of the short side between 480 and 800 and the long side no larger than 1333. The batch size is 16. We use learning rate $5e - 4$. The layerwise decay rate is 0.75, and the drop path rate is 0.2. We train the network with the $1\times$ schedule: 12 epochs with the learning rate decayed by $10\times$ at epochs 9 and 11. We do not use multi-scale testing. The Mask R-CNN implementation follows MMDetection.

**Scalability.** We scale the model size including ViT-B/16 and ViT-L/16 with 300 epoch on ImageNet1K. For ViT-B/16, in pretraining, we use 40 warmup epoch, $1.5e - 4$ base learning rate, 0.05 weight decay, and 4096 batch size. In fine-tuning, we use $5e - 4$ as base learning rate with 0.65 layerwise decay. The batch size is 1024, warmup epoch is 5, and weight decay is 0.05. The drop path is set to 0.1. For ViT-L/16, in pretraining, we use 30 warmup epoch, $5e - 6$ base learning rate. Due to the limited resource, we only use 1024 batch size. The weight decay is 0.05. In fine-tuning, we search from three learning rate $1e - 3$, $1.1e - 3$, and $1.2e - 3$ with 0.75 layer decay. We set the drop path as 0.2. The batch size is set to 1024 as well. Hence, the performance of ViT-L may not significantly outperforms MAE. We also scale the training time including 800 epoch and 1600 epoch. For 800 epoch ViT-B/16, in pretraining we use use 40 warmup epoch, $1e - 5$ base learning rate, 0.05 weight decay, and 4096 batch size. In fine-tuning, we use $5e - 4$ base learning rate, 0.65 layerwise decay, 0.05 weight decay. We set batch size to 1024, warmup epoch to 5, and drop path to 0.12. For 1600 epoch ViT-B/16, in pretraining we use use 40 warmup epoch, $5e - 6$ base learning rate, 0.05 weight decay, and 4096 batch size. As for fine-tuning, we use $5e - 4$ base learning rate, 0.65 layerwise decay, 0.05 weight decay. We set batch size to 1024, warmup epoch to 5, and we search from two drop path 0.1 and 0.12. For optimization, we use the same optimizer as MAE for both pretraining and fine-tuning.

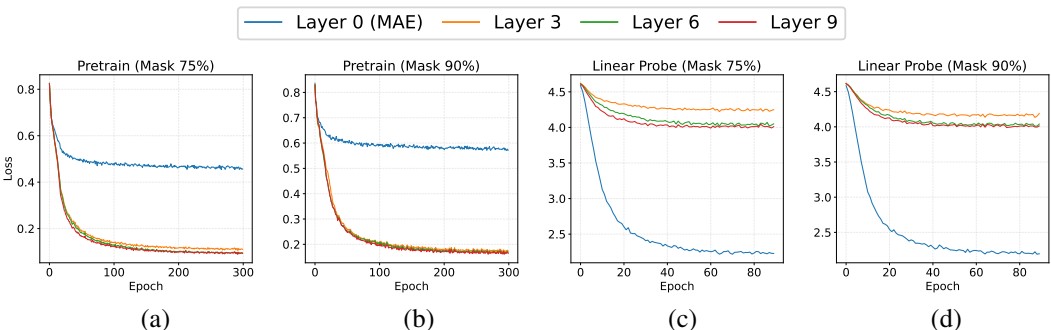

Figure 9: Loss curve of pretraining and linear probing with masking at 75% and 90% on the training tasks. We also illustrate the curves of finetuning in Appendix.

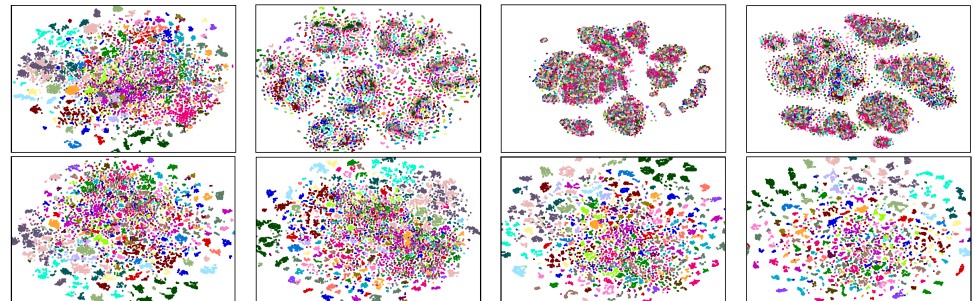

Figure 10: Visualization of the latent representations of the patches before (the first row) and after finetuning (the second row). Four columns from left to right represent encoders ($j = 0, 3, 6, 9$), respectively. Mask ratio $\eta_2$ is 0.9.

## A.2 ONE-SHOT MASKING

### A.2.1 REPRESENTATION DISCRIMINATION

We analyze the discrimination of output representation from pretrained model masked at different postions ($j = 0, 3, 6, 9$) from two aspects inclduing loss curve and representation visualization.

**Loss curve.** During pretraining, as depicted in Fig 9(a) and (b), masking at the beginning makes the optimization more challenging. This difficulty forces the encoder to learn more clues from visible patches. Consequently, in the linear probing phase (with fixed parameters), as shown in Fig 9(c) and (d), the encoder at $j = 0$ is more easily optimized compared with others. This implies that the representations encoded by the fixed encoder at $j = 0$ are relatively more distinguishable.

**Representation visualization.** To further verify this finding, we follow the approach of CAE (Chen et al., 2022b) and visualize the latent representations of patches from randomly sampled images from the ADE20K dataset in a 2D space using t-SNE (Van der Maaten & Hinton, 2008), as illustrated in Fig 10. We adopt t-SNE (Van der Maaten & Hinton, 2008) to visualize the high-dimensional patch representations output from our pretrained encoder on ADE20K (Zhou et al., 2019). ADE20K has a total of 150 categories. For each patch in the image, we set its label to be the category that more than half of the pixels belong to. We collect up to 200 patches for each category from sampled 500 images. In the first row, the latent representations of the encoder at $j = 0$ are clustered to some degree for different categories, while the encoders at $j = 3, 6, 9$ fail to achieve such clustering. Additionally, in the second row, finetuning causes the representations of different categories to scatter while those of the same category cluster together, thereby significantly enhancing the performance of pretrained encoders.

### A.2.2 ATTENTION MAP VISUALIZATION

To figure out what fine-tuning brings to encoders at $j = 0, 3, 6, 9$, we visualize the attention maps averaged over attention heads between the class token and the patch tokens in the last layer of ViT, as shown in Fig 11. It can be observed that fine-tuning narrows the attention scope of the

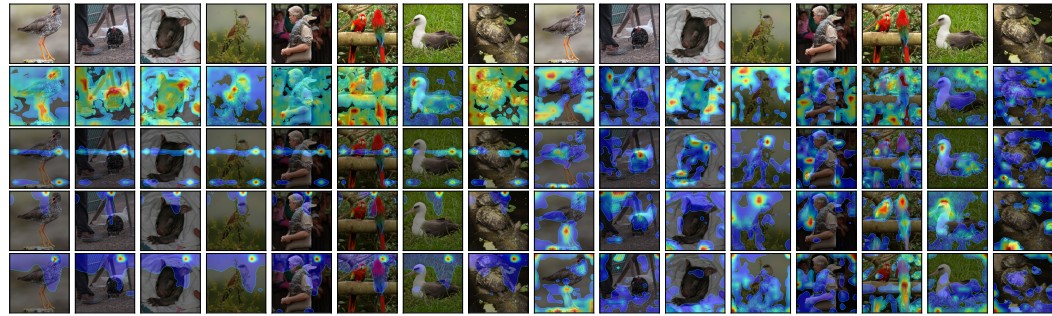

Figure 11: Visualization of the mean attention map of all heads in the last block of ViT before (1-8 columns) and after fine-tuning (9-16 columns). The region inside the blue contour is obtained by thresholding the attention weights to keep 50% of the mass. These images are randomly sampled from the ImageNet100 val set. The last four rows represent encoders ($j = 0, 3, 6, 9$), respectively.

encoder at $j = 0$, potentially removing some noise factors. In contrast, fine-tuning remarkably expands the attention field of encoders at $j = 3, 6, 9$, involving more information. Similar results can also be observed for each attention head in Fig 12 and Fig 13. It can be observed that fine-tuning narrows the attention scope of the encoder at $j = 0$, potentially removing some noise factors. In contrast, fine-tuning remarkably expands the attention field of encoders at $j = 3, 6, 9$, involving more information.

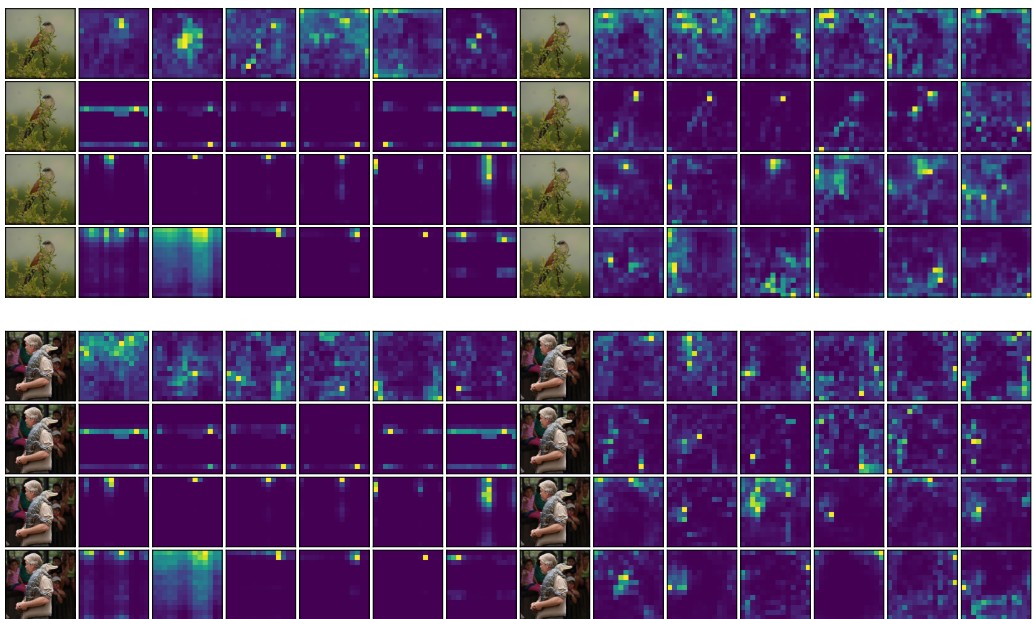

Figure 12: Visualization of the attention map of six heads in the last block of transformer encoder before and after finetuning. Four rows represents encoders (0-th, 3-th, 6-th, and 9-th) respectively.

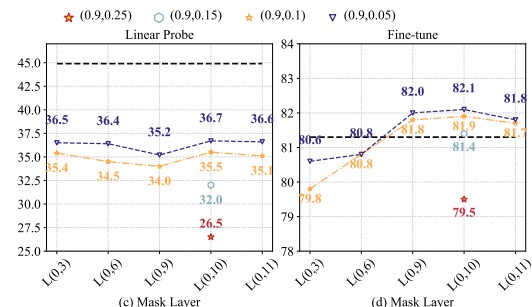

Figure 13: Visualization of the attention map of six heads in the last block of transformer encoder before and after finetuning. Four rows represents encoders (0-th, 3-th, 6-th, and 9-th) respectively.

## A.3 TWO-SHOT MASKING

### A.3.1 RESULTS OF TWO-SHOT MASKING

For $\eta_1 = 0.9$, in Fig 14 (c) and Figure 14 (d), using a smaller $\eta_2$, *e.g.*, 0.1 and 0.05, helps model obtain superior performance compared to $\eta_2 = 0.25$ and 0.15 in both linear probing and fine-tuning. We also find that the performance of two-shot masking is inferior to the baseline for linear probing. This is in line with expectation as $\eta_1$ is considerably large, resulting in quite few patches (clues) left. The second shot masking further eliminates the visible patches, making it more challenging to reconstruct the missing information. However, as opposed to linear probing, one can see that although $\eta_1 = 0.9$ is quite large, our two-shot masking still shows potential to outperform the baseline in fine-tuning, especially at $L(0, 10)$.

Figure 14: Results of two-shot masking on ViT-S/16. The dash line is the one-shot baseline (MAE).

### A.3.2 MORE VISUALIZATION

We first visualize We first leverage Centered Kernel Alignment (CKA) to analyze the layer representation similarity across pretrained models. As illustrated in Fig 15, we visualize the layer representation similarity between several two-shot masking pre-trained models and baseline $(0, 0.9)$ as heatmaps. We can see that the representation varies from that of baseline, similar to two-shot models $\eta_1 = 0.75$.

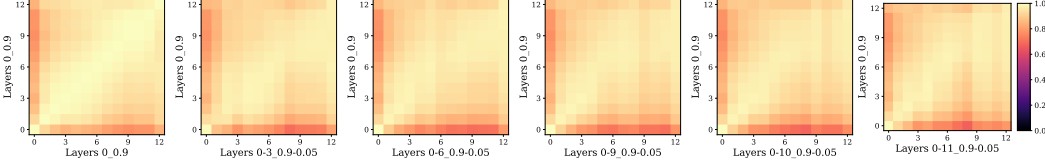

Figure 15: Layer representation similarity between pretrained two-shot masking model and baseline.

Then we visualize the attention distance and attention entropy over different two-shot models and baselines in Fig 16 ($\eta_1 = 0.75$) and Fig 17 ($\eta_1 = 0.9$). We see that the second masking decreases the attention distance and entropy for all two-shot models no matter where the position of the second masking is.

We also present the attention distance and attention entropy before/after fine-tuning for two-shot model variants ($\eta_1 = 0.75$) shown in Fig 18. Compared to pretraining, fine-tuning decreases the attention distance and entropy in low layer and also elevates attention distance in high layer for all models.

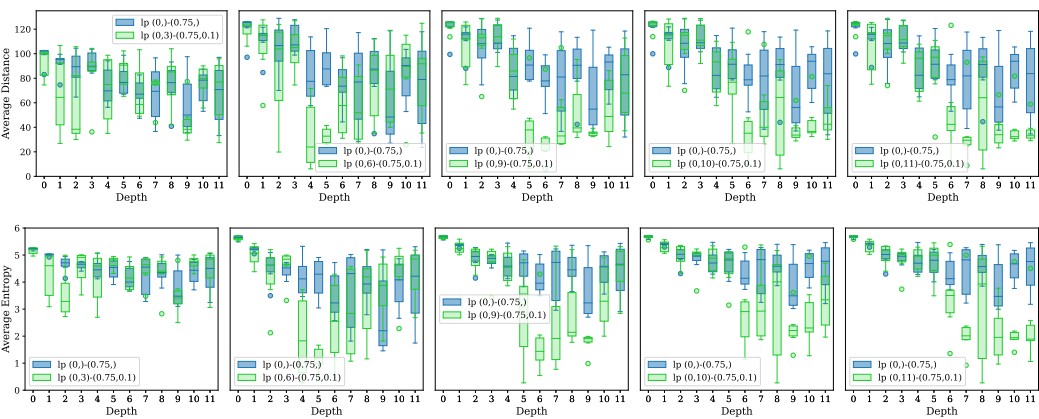

Figure 16: Comparison of two-shot masking and baseline model ($\eta_1 = 0.75$) on attention distance and attention entropy.

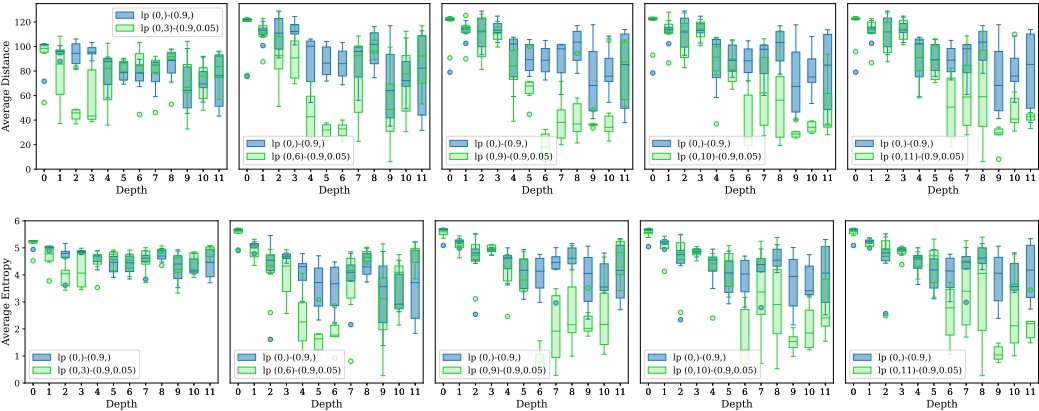

Figure 17: Comparison of two-shot masking and baseline model ($\eta_1 = 0.9$) on attention distance and attention entropy.

Finally, we compare the attention distance and entropy between baseline and two-shot model variants after fine-tuning ($L(0, 3/6/9/10/11; 0.75, 0.1)$) in Fig 19. We see that two-shot model variants have similar attention distance and entropy in high layers while more concentrated and lower attention distance and entropy in low and middle layers.

### A.4 THREE-SHOT MASKING

We present the three-shot result in Tab 7. The "Equal interval" strategy refers to equally spaced masking positions, while the "Prefer front layer" indicates that the three-shot masking is performed in the early layers. The "Unbalanced interval" strategy selects the third masking position based on the best two-shot masking setting, which could be close to either the first or second masking position.

Among different strategies, we find our three-shot masking method $(\eta_1, \eta_2, \eta_3) = (0.75, 0.1, 0.1)$ yielded the best results. This highlights the superiority of our step-by-step strategy, which exhibits a resemblance to the greedy algorithm.

### A.5 ROBUSTNESS ANALYSIS

Considering Conditional-MAE suffers extra masking, it should be intuitively more robust than MAE. To verify it, we use a fine-tuned model to conduct two kinds of perturbation schemes, *i.e.*, occlusion and shuffling, aiming to simulate the real circumstances. For occlusion, we randomly mask half of the patches following (Zhou et al., 2021) before inputting the model. For shuffling, we randomly

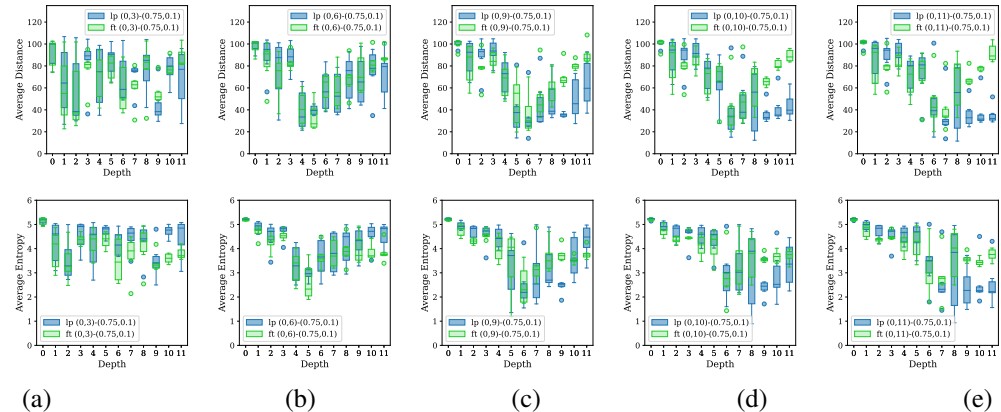

Figure 18: Comparison of two-shot model variants $L(0, 3/6/9/10/11; 0.75, 0.1)$ and baseline model $L(0; 0.75)$ on attention distance and attention entropy before/after fine-tuning. Lp means pretrained model. Ft means fine-tuned models.

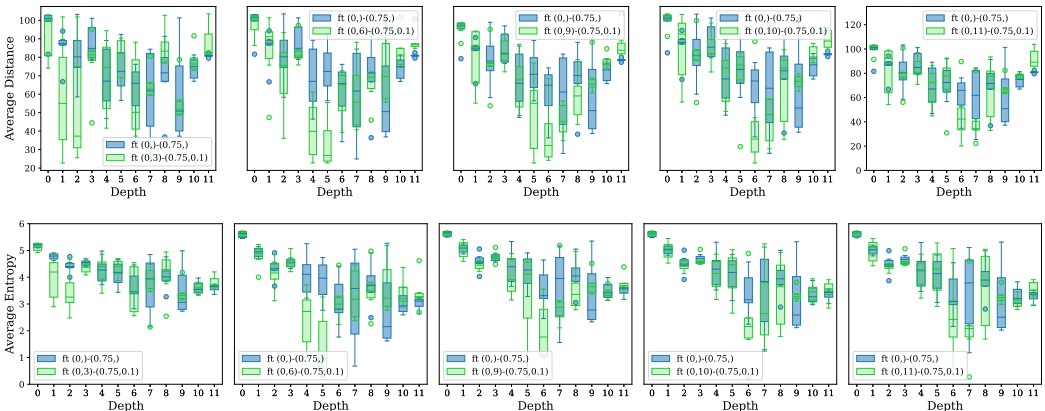

Figure 19: Comparison of two-shot model variants ($L(0, 3/6/9/10/11; 0.75, 0.1)$) and baseline model on attention distance and attention entropy.

shuffle the patches as well. As presented in Tab 9, compared to Tab 5, Conditional-MAE suffers less performance drop than MAE, indicating more excellent robustness.

## A.6 MORE RELATED WORK

**Masking in generation modeling.** Chang *et al*. introduce MaskGIT (Chang et al., 2022), which employs a bidirectional transformer decoder and is capable of learning to predict randomly masked tokens via attending to tokens in all directions during training. When inference, MaskGIT first generates all tokens of an image and then refines the generated image iteratively based on the previous generation. Recently, Chang *et al*. propose Muse (Chang et al., 2023) and train it to predict randomly masked image tokens given the text embedding extracted from a pre-trained large language model (LLM). Leveraging LLM enables Muse to understand fine-grained language, translate to high-fidelity image generation, etc. Moreover, Muse directly enables inpainting, outpainting, and mask-free editing without the need to fine-tune or invert the model. Li *et al*. (Li et al., 2023c) propose to use semantic tokens learned by a vector-quantized GAN at inputs and outputs and combine this with masking to unify representation learning and image generation. Bandara *et al*. propose an adaptive masking strategy called AdaMAE (Bandara et al., 2023). AdaMAE samples visible tokens based on the semantic context using an auxiliary sampling network and empirically demonstrates the efficacy. Xiao *et al*. introduce a simple yet effective adaptive masking over masking strategy called AMOM (Xiao et al., 2023) to enhance the refinement capability of the decoder and make the encoder optimization easier.

Table 7: Our step-by-step three-shot masking compared to others three-shot masking strategies.

| Different three-shot masking strategy | $i, j, k$ | $\eta_1, \eta_2, \eta_3$ | FT |
|---|---|---|---|
| | 0, 4, 8 | 0.5, 0.5, 0.5 | 66.42 |
| Equal interval | 0, 5, 10 | 0.5, 0.5, 0.5 | 67.2 |
| | | 0.75, 0.25, 0.25 | 73.5 |
| | | 0.75, 0.25, 0.1 | 77.7 |
| | | 0.75, 0.1, 0.1 | 80.9 |
| | 0, 6, 11 | 0.75, 0.25, 0.1 | 77.7 |
| | | 0.75, 0.1, 0.1 | 80.9 |
| Prefer front layers | 0, 3, 6 | 0.5, 0.5, 0.5 | 64.1 |
| | | 0.75, 0.25, 0.1 | 77.1 |
| | | 0.75, 0.1, 0.1 | 80.5 |
| | 0, 2, 4 | 0.75, 0.1, 0.1 | 80.3 |
| | 0, 1, 2 | 0.75, 0.1, 0.1 | 81.4 |
| Unbalanced interval | 0, 3, 10 | 0.5, 0.5, 0.5 | 64.7 |
| | | 0.75, 0.25, 0.1 | 78.1 |
| | | 0.75, 0.1, 0.1 | 81.3 |
| | 0, 2, 10 | 0.75, 0.1, 0.1 | 81.4 |
| | 0, 1, 10 | 0.75, 0.1, 0.1 | 81.8 |
| | 0, 9, 10 | 0.75, 0.1, 0.1 | 81.7 |
| | 0, 8, 10 | 0.75, 0.1, 0.1 | 81.6 |
| Our three-shot | 0, 10, 11 | 0.75, 0.1, 0.1 | **81.9** |

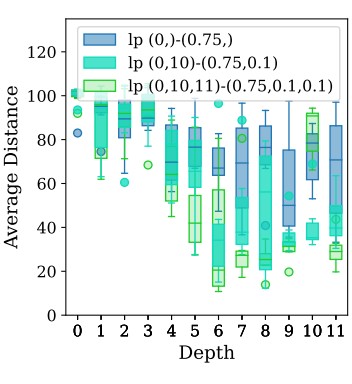
(a) Attention Distance

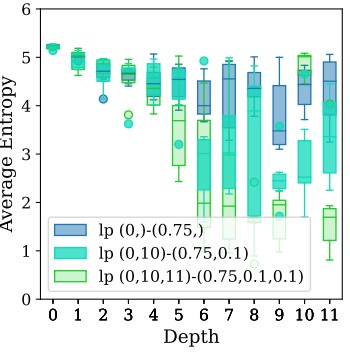
(b) Attention Entropy

Figure 20: Comparison on attention distance and entropy among one-shot, two-shot, and three-shot masking.

Table 8: Comparisons on fine-grained datasets among one-shot, two-shot, and three-shot masking.

| Dataset | $L(0; 0.75)$ | $L(0, 10; 0.75, 0.1)$ | $L(0, 10, 11; 0.75, 0.1, 0.1)$ |
|---|---|---|---|
| ImageNet100 | 82.5 | 84.6 (+2.1) | 81.9 (-0.6) |
| Flower102 | 34.7 | 37.3 (+2.6) | 35.1 (+0.4) |
| Standford Dog | 51.6 | 54.3 (+2.7) | 52.2 (+0.6) |
| CUB-200 | 48.2 | 51.1 (+2.9) | 48.7 (+0.5) |

Table 9: Robustness analysis (occlusion and shuffling) of Conditional-MAE and MAE with four classification datasets.

| Model | occlusion | | | | shuffling | | | |
|---|---|---|---|---|---|---|---|---|
| | DTD | CF10 | CF100 | Tiny | DTD | CF10 | CF100 | Tiny |
| MAE | 56.3 | 71.6 | 48.4 | 49.9 | 47.7 | 68.8 | 45.6 | 42.9 |
| Conditional-MAE | **57.8** | **72.8** | **49.5** | **51.2** | **49.1** | **70.2** | **47.1** | **44.0** |

