# OpenReview forum: "An Empirical Study of Multiple Masking in Masked Autoencoder"
_ICLR.cc/2025/Conference — ICLR 2025 Conference Withdrawn Submission_

### Official Review · Reviewer_9sBm · 2024-10-30

**Soundness:** 3
**Presentation:** 3
**Contribution:** 3
**Rating:** 5
**Confidence:** 4

**Summary:**

This paper introduces a framework named Conditional MAE, designed to explore the effects of applying multiple rounds of masking within masked autoencoders. The goal is to systematically analyze how these successive masking stages impact training optimization and model performance. Conditional MAE adapts each round of masking based on the previous unmasked representations, enabling more adaptable masking strategies. The study assesses one-shot, two-shot, and three-shot masking, revealing that multiple rounds of masking can induce locality bias, which may improve performance in specific tasks. Experiments with generalization to different architecture are also included.  The authors demonstrate that Conditional MAE outperforms the standard MAE in some downstream applications.

**Strengths:**

1. This paper studies the empirical behavior of multi-round masking in the encoder and provides extensive experimental results.
2. Some empirical analysis are provided on why/how two shot masking benefits MAE with visualizations.
3. The paper is well-paced and written and is easy to follow. The content is clear.

**Weaknesses:**

1. Scaling experiments of Conditional-MAE offers trivial improvements (<0.3%) on ImageNet1K, while most of the existing explorations happened on a much smaller subset, ImageNet 100. My concern is that Conditional MAE will only offer limited benefits with large pretrain dataset.
2. Following point 1, transfer learning experiments are also conducted using ImageNet 100 pretrained model. I would like to see a L(0,10; 0.75,0.1) setting of ViT/B pretrained with 1600 epochs on ImageNet1k (from the scaling experiment) transfer learning results.
3. The only baseline studied in this paper is MAE. More baseline should be included.
4. The mask ratio conclusion from one-shot experiment is trivial, it is explored both empirically and theoretically in the previous works [1, 2]
5. Analysis on why more shot masking is hurting the performance of the model should be discussed, with more detailed discussion on "over-locality."


[1] Understanding Masked Autoencoders via Hierarchical Latent Variable Models, CVPR2023

[2] How Mask Matters: Towards Theoretical Understandings of Masked Autoencoders, NeurIPS 2022

**Questions:**

please refer to the Weakness section

---

### Official Review · Reviewer_NfNN · 2024-11-04

**Soundness:** 2
**Presentation:** 2
**Contribution:** 3
**Rating:** 5
**Confidence:** 3

**Summary:**

This paper systematically investigates the performance of multi-Masked AEs and their performance on downstream tasks. The authors introduced a series of MAE with multiple masks, named as Contidional MAE. They also investigates the mask selection strategy through experiments.

**Strengths:**

**1.** The experiments in this paper are complete and systematic for the position of those masks, mask ratios, and other settings.

**2.** The study in this paper shows very clearly how multiple Masks affect the performance of MAEs when ViT is used as a backbone, which is instructive for the future development of MAEs and multimodal MAEs.

**Weaknesses:**

**1.** This paper lacks theoretical analysis.
The authors could try to explain the effect of multiple masks on MAE from the point of view of information bottleneck theory, or the trade-off between the number of masks and reconstruction loss, etc. These theoretical analyses should be corresponded with experiments to make the paper more solid.

**2.** The authors would do well to check for article consistency. The authors claim that they analysed robustness, but no such subsection appeared in the experiment.

**Questions:**

How does the network performance change if there are more than 3 masks, but the probability of each mask is low? It is hoped that the authors can provide more extensive experimental results.

---

### Official Review · Reviewer_n9j9 · 2024-11-08

**Soundness:** 3
**Presentation:** 3
**Contribution:** 1
**Rating:** 5
**Confidence:** 3

**Summary:**

This paper studies how multiple masking can be helpful in MAE. A framework called Conditional MAE is proposed to enable the analysis. Extensive experiments are conducted to support the takeaways and findings, such as multiple masking introduces locality bias.

**Strengths:**

- The related works are well covered.
- The methodology is clearly explained with nice figures and reasonable formal equations.
- Extensive experiments are conducted to support most of the claims and findings.

**Weaknesses:**

- I don’t see a clear motivation on why we should pay much attention on multiple masking. The third paragraph of the Introduction tries to motivate the research question. However, it only discusses how the three previous works (UnMAE, VideoMAE v2, A2MIM) adopt the idea of multiple masking differently (than the proposed Conditional MAE?). Therefore, the research question does not arise “naturally”. In my opinion, for the question to arise naturally, we need to demonstrate that multiple masking is indeed an effective method first, then the readers will be more interested as there is a lack of understanding of it.
- The key takeaways are more like a recipe supported by experiments than an “in-depth and comprehensive analysis”. Only “the second masking brings locality bias into the model and helps capture low-level features” serves as a conclusion from analysis that deepens our understanding of multiple masking. However, I think this finding is pretty intuitive and not so valuable. I refer to [a] as a sample work of analyzing and understanding MIM. On the other hand, recipes can be very meaningful when the topic is significant enough (e.g., [b]), but multiple masking may not be one of such topics.
- Some of the experiments are not very extensive. For example, only ViT-B and ViT-L are used to show the scalability. If the computational resources are limited, the authors may consider ViT-S and ViT-tiny, or some other efficient variant of ViTs.

[a] Zhang et al. How Mask Matters: Towards Theoretical Understandings of Masked Autoencoders. NeurIPS, 2022.

[b] Mo et al. When Adversarial Training Meets Vision Transformers: Recipes from Training to Architecture.

Minor:

- Line 071: missing dot.

**Questions:**

- Does “11” in equation (6) come from the total number of layers, i.e., 12, in the MAE encoder? If so, I think you might consider avoiding using such a specific number here, or you should at least explain it.

- Although you tried to explain the name Conditional MAE in Sec 2.2, I still think the name may not be appropriate as the core idea and aim of this framework is about multiple masking. Do you think something like Multi-MAE could be better?

---

### Note · Authors · 2024-11-20

I have read and agree with the venue's withdrawal policy on behalf of myself and my co-authors.